# Is Antioxidant Therapy a Useful Complementary Measure for Covid-19 Treatment? An Algorithm for Its Application

**DOI:** 10.3390/medicina56080386

**Published:** 2020-07-31

**Authors:** María Elena Soto, Verónica Guarner-Lans, Elizabeth Soria-Castro, Linaloe Manzano Pech, Israel Pérez-Torres

**Affiliations:** 1Immunology Department, Instituto Nacional de Cardiología “Ignacio Chávez”, Juan Badiano 1, Sección XVI, Tlalpan, México City 14080, Mexico; mesoto50@hotmail.com; 2Physiology Department, Instituto Nacional de Cardiología “Ignacio Chávez”, Juan Badiano 1, Sección XVI, Tlalpan, México City 14080, Mexico; veronica.guarner@cardiologia.org.mx; 3Vascular Biomedicine Department, Instituto Nacional de Cardiología “Ignacio Chávez”, Juan Badiano 1, Sección XVI, Tlalpan, México City 14080, Mexico; elizabethsoria824@gmail.com (E.S.-C.); loe_mana@hotmail.com (L.M.P.)

**Keywords:** COVID-19, antioxidant therapy, ACE2 receptor, sepsis, SARS-CoV-2

## Abstract

Severe Acute Respiratory Syndrome Coronavirus 2 (SARS-CoV-2) causes the corona virus disease-19 which is accompanied by severe pneumonia, pulmonary alveolar collapses and which stops oxygen exchange. Viral transmissibility and pathogenesis depend on recognition by a receptor in the host, protease cleavage of the host membrane and fusion. SARS-CoV-2 binds to the angiotensin converting enzyme 2 receptor. Here, we discuss the general characteristics of the virus, its mechanism of action and the way in which the mechanism correlates with the comorbidities that increase the death rate. We also discuss the currently proposed therapeutic measures and propose the use of antioxidant drugs to help patients infected with the SARS-CoV-2. Oxidizing agents come from phagocytic leukocytes such as neutrophils, monocytes, macrophages and eosinophils that invade tissue. Free radicals promote cytotoxicity thus injuring cells. They also trigger the mechanism of inflammation by mediating the activation of NFkB and inducing the transcription of cytokine production genes. Release of cytokines enhances the inflammatory response. Oxidative stress is elevated during critical illnesses and contributes to organ failure. In corona virus disease-19 there is an intense inflammatory response known as a cytokine storm that could be mediated by oxidative stress. Although antioxidant therapy has not been tested in corona virus disease-19, the consequences of antioxidant therapy in sepsis, acute respiratory distress syndrome and acute lung injury are known. It improves oxygenation rates, glutathione levels and strengthens the immune response. It reduces mechanical ventilation time, the length of stay in the intensive care unit, multiple organ dysfunctions and the length of stay in the hospital and mortality rates in acute lung injury/acute respiratory distress syndrome and could thus help patients with corona virus disease-19.

## 1. Introduction

In December 2019 there was an outbreak of a novel disease called Corona Virus Disease-2019 (COVID-19) caused by the Severe Acute Respiratory Syndrome Coronavirus 2 (SARS-CoV-2). The virus consists of a positive-sense single-stranded RNA surrounded by a membrane in which proteins are inserted. It is a variant of the SARS-CoV-1 with which it shares 82% of the genome sequence and of the Middle East Respiratory Syndrome Coronavirus (MERS-CoV) with a 50% genome sequence homology [1]. The RNA genome codes for non-structural and structural proteins. It codifies for four groups of proteins—(1) viral envelope proteins (E) that include the spike structural glycoprotein (S) with prominent projections on the surface and which is crucial for virus attachment to the host and determines tropism and transmission capacity [2], the nucleocapsid (N) protein which determines the shape of the viral particle and binds to the nucleocapsid [3] and the matrix protein (M) which helps in the RNA binding to the host transcription complex participating in viral replication [4], (2) proteases nsp3 and nsp5, (3) a RNA-dependent RNA polymerase (RdRp) which is a non-structural protein that increases the rate of replication of RNA from templates [5] and (4) a set of 4 or more 3′ conterminal sub-genomic mRNAs which are a replicase whose genes are composed of two overlapping open reading frames [5]. The ability of this virus and other coronaviruses to mutate makes them zoonotic pathogens of concern since it facilitates the transmission between animals and humans [6]. The evolutionary analysis of the virus shows that the SARS-CoV-2 is very similar to the isolated Bat coronavirus RaTG13 (GenBank No.MN996532), with 96.2% nucleotide homology in the whole genome [7] and therefore, this virus might have evolved from the bat coronavirus. The infection by this virus started in Hubei, in the city of Wuhan, in China and rapidly spread throughout this country and the world. The World Health Organization (WHO) ranked it as a world pandemic in March.

The COVID-2019 disease results in a severe pneumonia that leads to pulmonary alveolar collapse in a few h and that leads to the cessation of oxygen exchange. Figure 1 shows the ultrastructure of the middle zone of a bronchiole in which viral particles can be observed. However, in comparison Figure 2 shows the ultrastructure of the middle zone of a bronchiole without viral particles.

The virus has an incubation period of 2 to 10 days and the clinical spectrum of the disease ranges from a severe respiratory failure to an asymptomatic infection. Patients with the severe form of COVID-19 show a high respiratory frequency (≥30/min), dyspnea, abnormal partial pressure of arterial oxygen to fraction of inspired oxygen ratio (of <300), low blood oxygen saturation (≤93%), and/or lung infiltrates (>50%). Patients develop septic shock, respiratory failure, and/or multiple organ failure [8]. There is elevated lymphopenia, lactate creatinine and kinase dehydrogenase and a cytokine storm characterized by increased concentrations of IL-5, IL-1β, IL-7, IL-8, IL-10, IL-9, IL-15, IL-12p70, PDGF, VEGF, GCSF, FGF, IFN-γ, GMCSF, IP-10, MCP-1, MIP-1A, MIP-1B and TNF-α [9]. This condition finally leads to total organic failure. The inflammatory response could be triggered by free radicals. Oxidative stress (OS) mediates the activation of NFkB inducing in turn the transcription of genes promoting cytokine production. Release of these cytokines, in turn, enhances the inflammatory response. Oxidizing agents come from the phagocytic leukocytes such as neutrophils, monocytes, macrophages and eosinophils invading the tissue. Free radicals promote cytotoxicity thus injuring cells. They degrade essential cellular components and alter the protease/antiprotease balance in the tissue interstitium [10].

In this paper we discuss the mechanism of action of the virus and the way in which this mechanism correlates with comorbidities that increase the death rate in patients. We also discuss the currently proposed therapeutic measures and propose the use of antioxidant drugs as an adjuvant to improve the conditions of infected subjects with the severe form of the COVID-19.

## 2. Inter-Relations between the Virus and the Host Cell Interaction of the Mechanism with the Comorbidities Found in the Severe Forms of COVID-19

The transmissibility and pathogenesis of the viruses depend mainly on receptor recognition by the host, cleavage by a protease and membrane fusion. Entry depends on binding of the surface unit, S1, of the S protein to the angiotensin-converting enzyme 2 (ACE2) receptor, which facilitates viral attachment to the surface of target cells and the serine protease TMPRSS2 for S protein priming which entails S protein cleavage at the S1/S2 and the S2 site and allows fusion of viral and cellular membranes. This process is driven by the S2 subunit [11]. The ACE2 protein is present in the circulation as a soluble molecule and it binds to the lung, liver, kidney, brain and heart [12]. It is an aminopeptidase that binds to the membrane and plays a vital role in the immune and cardiovascular systems of the host. It is involved in the development of hypertension, cardiovascular disease (CVD), metabolic syndrome (MS) and diabetes [13]. The specific binding mechanisms of the virus are uncertain. Figure 3 describes the relationship between SARS-CoV-2, ACE2 and the cardiometabolic disorders.

The ACE2 receptor-related signaling pathways play an important role in several pathologies that are considered as comorbidities that increase the death rate of COVID-19 [14]. Patients with hypertension and CVD have a high expression of this receptor and this might explain, in part, why symptoms are more pronounced in these patients and in related diseases including MS, diabetes, hypertension and hepatic diseases [14].

In a study that included 1099 subjects with confirmed COVID-19, it was found that 173 that had the severe form of the disease also had comorbidities such as coronary heart diseases (5 (8%)), diabetes mellitus (16 (2%)), cerebrovascular disease (2 (3%)) and hypertension (23 (7%)) [15]. COVID-19 may cause more alterations in the modified pathways of the ACE2 receptor that are present in patients [16]. The interrelation between the pathologies which comprise the MS and COVID-2 has also been demonstrated by several studies [17].

### 2.1. COVID-19 and CVD (Hypertension and Myocardial Injury)

Hypertension and myocardial injury are often treated with inhibitors of the renin-angiotensin-aldosterone system (RAAS) which may elevate the levels of the ACE2 receptor. Recent reports suggest that myocardial injury is related with the SARS-CoV-2. Figure 4 shows ultrastructure features of a heart sample from a patient with COVID-19 postmortem showing the presence of the virus. However, in comparison the Figure 5 shows the ultrastructure of a heart sample as control subject without viral particles.

Patients with severe symptoms have frequently complications including acute myocardial injury which is associated with ACE2 receptor alterations [18]. Another proposed mechanism of myocardial injury includes a cytokine storm that is initiated by a loss of the equilibrium between type 1 and type 2 T helper cells responses, respiratory malfunctioning and hypoxemia as a consequence of COVID-19 giving rise to myocardial cell damage [18].

Blood pressure (BP) levels are also more elevated in COVID-19 patients treated in the ICU in contrast to patients that did not require the entry into the ICU. ACE2 inactivates Ang II and provides Ang 1–7, which vasodilates vessels by activating the MAS receptor, thus negatively regulating RAAS [19]. Furthermore, the AT1R antagonists which are commonly used to reduce BP, elevate the ACE2 receptor expression in the heart in about three times [20]. In hypertensive patients that received the AT1R antagonist olmesartan, higher urinary ACE2 levels were observed [21]. Therefore, the blockade of AT1R for long periods may result in ACE2 receptor up-regulation, rendering the cells of these patients more susceptible to COVID-19 due to an increased number of cellular binding sites [22].

### 2.2. COVID-19 and Liver Diseases

COVID-19 severity is associated with liver diseases such as cirrhosis, non-alcoholic fatty liver disease, alcohol-related liver disease and chronic viral hepatitis and the presence of the virus in liver has been confirmed in pathological studies in patients with SARS [23]. Patients having COVID-19 showed liver comorbidities and in these patients there were altered levels of AST and ALT. Patients also showed diminished serum albumin levels and elevated serum bilirubin levels [24]. The direct viral infection of liver cell may be the cause of liver damage [25]. There is also an elevation of gamma glutamyl transferase, which is a diagnostic biomarker for cholangiocyte injury [24]. This suggests that ACE2 receptor expression may be enhanced in cholangiocytes and SARS-CoV-2 might bind to ACE2-positive cholangiocytes to alter the control of the liver function [26].

Clinical investigations of COVID-19 have also found an increased AST level, prothrombin time prolongation and hypoproteinemia. The direct infection of the liver was confirmed by the existence of SARS viral nucleic acids in the liver. Moreover, apoptosis and mitoses, together with atypical features including ballooning of hepatocytes, acidophilic bodies and fibrosis or lobular activities without fibrin deposition, were found in percutaneous liver biopsies of SARS. The SARS-associated hepatotoxicity may be present when there is viral hepatitis or it may appear as an effect of the toxicity of consumed drugs such as elevated doses of antibiotics, antiviral medications and steroids or to an increased reaction of the immune system [13]. Large numbers of viruses in the liver were found in autopsies performed on SARS patients [27]. This suggests that the ACE2 receptor, is present in liver endothelial cells and the presence of these receptors render this organ as a potential target for SARS-CoV-2 [28]. Figure 6 shows the liver ultrastructure features of a patient with COVID-19 postmortem. However, in comparison the Figure 7 shows the ultrastructure of a liver sample s control subject without viral particles.

The ACE2 expression in bile duct cells is even more elevated than in hepatocytes and its levels of expression resemble those found in lung alveolar type 2 cells. Epithelial cells from the bile duct importantly participate in the regeneration of the liver and the immune response. Thus, liver injury in COVID-19 subjects may result from damage to bile duct cells [29].

The liver biopsy of a patient with COVID-19 also showed steatosis and liver injury [30]. The infection with COVID-19 and the development of the disease are accompanied by the presence of many hepatic alterations, which may result from the effect of the virus on the liver or from side effects of the medications [31]. In addition, liver injury may also result from immune-mediated inflammation, with a cytokine storm and pneumonia-associated hypoxia. This may also lead to liver failure in critically ill patients with COVID-19 [29].

### 2.3. COVID-19 and the Respiratory System and Its Association with the Inflammatory Process

SARS-CoV-2 is found in swabs taken from the nose and throat of patients who are suspected to have or are affected by COVID-19 [32]. In this sense, when SARS-CoV-2 is present in the lower and upper respiratory tract, it results in mild or severe acute respiratory syndrome. The linking of SARS-CoV-2 to the Toll Like Receptor (TLR) results in the liberation of pro-IL-1β and there is then the liberation of pro-inflammatory cytokines, such as interleukin IL-6 and IL-1β. This activates the inflammasome and leads to the release of active mature IL-1β which causes lung inflammation, fibrosis and fever [32].

In COVID-19, there is a decreased lymphocyte count, elevated leukocyte count and neutrophil-lymphocyte-ratio and lower percentages of basophils, eosinophils and monocytes. Severe cases usually have high levels of inflammatory cytokines and of biomarkers related to infection and the number of T lymphocytes CD3+ and CD4+ were significantly decreased and were more hampered in severe cases. This suggests, a deregulated immune system but no important change in CD8+ and B cells. Cytokines and chemokines importantly participate in immunopathology and immunity during viral infections [32]. SARS-CoV-2 may have direct cytopathic effects and may lead to evasion of host immune responses which play a major role in the severity of COVID-19. The first line of defense against viral infections is a rapid and regulated innate immune response; however, when the immune response is uncoordinated, it will result in an excessive inflammation which may even cause death [33].

The reduction of eosinophils could be associated to a stress response in the acute lung injury caused by the virus that leads as a consequence to the suppression of the release of eosinophils by the bone marrow caused by glucocorticoid secretion. Therefore, low eosinophils might suggest COVID-19 progression and their increasing numbers might indicate recovery [34]. In patients infected with SARS-CoV-2 there are significantly higher levels of D-dimer, PKC and procalcitonin and this was related to severe patients in contrast to non-severe patients [24].

The synthesis and liberation into the blood stream of extrathyroidal procalcitonin is increased during bacterial infections and is maintained by IL-6, IL-1β and TNF-α. Procalcitonin is a mediator in the systemic inflammatory response syndrome which is a well-known characteristic of multi-organ failure, septic shock and severe sepsis [35]. However, the production of procalcitonin-biomarker is decreased by INF-γ, whose concentration elevates during viral infections. Therefore, procalcitonin value is sustained within the normal levels in patients with SARS-CoV-2 non-complicated infection and its elevation may indicate bacterial co-infection in patients developing the severe form of the disease [35].

### 2.4. SARS-Cov-2 Associated with Oxidative Stress and Inflammatory Process

In chronic obstructive pulmonary disease, acute lung injury (ALI) and acute respiratory distress syndrome (ARDS) there is an increase in reactive oxygen species (ROS) and reactive nitrogen species (RNS) [36]. These events are associated with the increased release of pro-inflammatory mediators such as IL-6, IL-8 and TNF-α by bronchial epithelial cells and alveolar macrophages [37], which may then activate neutrophils and macrophages, resulting in alveolar wall destruction and the collapse of small airways [38]. These changes may induce endothelial damage, pulmonary capillary hyper-permeability and pulmonary edema, resulting in the deterioration of pulmonary gas exchange [39]. Furthermore, in severe sepsis in ARDS, when there is life-threatening organ dysfunction caused by the host’s inadequate response to infection, the cardiovascular system increases the cardiac output and lowers peripheral resistance adopting a hemodynamic profile that leads to arterial dilatation [40]. An excessive drop in peripheral resistance or its prolonged time course may result in progressive hypotension that is refractory to catecholamines and may contribute to severe cardiovascular failure [41]. In different experimental models and in humans with severe septic shock, there is a high production and release of superoxide (O_2_^−^) and peroxynitrite (ONOO^−^) by different pathways that contributes to failure of the lungs, heart, brain and liver [42]. Despite limited clinical data, moderate and severe septic shock is developed in many viral diseases such as SARS-CoV, which may increase the production of ROS and RNS. Overproduction of these molecules is associated to an elevated expression of iNOS, NADP oxidases, cyclooxygenase 2 and xanthine oxidase that activate transcription factors such NFkB resulting in an exacerbated proinflammatory host response [43,44]. In addition, O_2_^−^ and ONOO^−^ participate as important mediators of the pro-inflammatory interleukin production. Moreover, these molecular species will continue to stimulate the production and release of more ROS and RNS which may interfere with mitochondrial respiration, since mitochondrial dysfunction is commonly induced in a septic shock environment [45]. Therefore, treatment with antioxidants can be a way to bypass the excessive inflammation associated to the background high oxidative state in COVID-19 patients.

## 3. Treatment of COVID-19

The main pharmacological treatments that are currently being used in the world against COVID-19 have shown relevance in different countries. Their therapeutic effects depend on the population genetics, age, gender and comorbidity conditions at the time of infection. The drugs that are being used may also have side effects which may have an impact on the outcome of the patient. A delay in the diagnosis may also alter the effect of the drugs. Moreover, during the evolution of an inflammatory disease triggered by an infection, the mechanisms of damage are interspersed, rendering difficult the decision of which drug is more important at a given time. However, during the therapeutic stage, attention must be paid to all of the factors that participate.

Given that there is a lack of a vaccine against this virus, a two-phase combined therapy strategy is required. The first phase is aimed to lower the viral load that is the source and origin of the chronic inflammatory condition leading to severe sepsis and multiple organ failure. The uncontrolled inflammatory condition might be decreased by the use of antioxidants. The second phase is aimed to reduce the sepsis condition, thereby reducing multiple organ failure. Since the two phases converge simultaneously, the therapy strategy should be selected avoiding subsequent collateral damage for the patient.

### Chloroquine and Hydroxychloroquine

Chloroquine has been proposed for COVID-19 treatment. This drug has been employed throughout the world for more than 70 years, being a component of the list of essential medicines of WHO. It is not expensive and has a well-established clinical safety profile. Chloroquine and hydroxychloroquine are nowadays employed against autoimmune disease and in the prophylaxis and treatment of malaria. These drugs also suppress the replication of some DNA and RNA viruses, such as SARS-CoV-1 [46].

Chloroquine is a form of quinine consisting of an acidotropic amine. It was synthesized in 1934 by Bayer and surged as an efficient replacement for natural quinine. Quinine is found in Cinchona trees that grow in Peru. In vitro, chloroquine is a versatile bioactive agent that has antiviral activity against RNA viruses including HIV, hepatitis A and C viruses, influenza A H5N1 virus, influenza A and B viruses, Chikungunya virus, Dengue virus, Zika virus, the rabies virus, poliovirus, Lassa virus, Nipah and Hendra viruses, Crimean-Congo hemorrhagic fever virus and the Ebola virus. It also inhibits the herpes simplex virus and the hepatitis B virus which are DNA viruses [46].

Chloroquine suppresses the entrance of viruses into host cells and inhibits glycosylation of viral proteins [47]. Chloroquine was effective in decreasing viral replication, in a concentration (EC) 90 of 6.90 μM that is reached with the common dosing of this drug, since it easily penetrates in tissues, such as the lung [47]. Chloroquine blocks viral infection by alkalinizing the endosomal pH in the phagolysosome and by blocking the glycosylation of cellular receptor of SARS-CoV. SARS-CoV-1A also has a pH-dependent mechanism of entering to target cells after its union to the DC-SIGN receptor. The activation that occurs in the acidic pH endosomes leads to fusion of the virus to the endosomal membranes leading to the liberation of its genome to the cytosol. When there is no antiviral drug, the virus is conducted to the lysosomes where there is an acidic pH which together with the activity of enzymes destroys the viruses, releasing the nucleic acid and the required enzymes for replication [48].

Chloroquine also alters the immune system functioning through activation of cell signaling and pro-inflammatory cytokine control [49]. Chloroquine suppresses phosphorylation of caspase-1 and of the p38 mitogen-activated protein kinase in THP-1 cells. Activation of cells via mitogen-activated protein kinase signaling is necessary for the viruses to replicate. In the model of HCoV-229 coronavirus, chloroquine inhibited the virus by suppression of p38 mitogen-activated protein kinase. Moreover, chloroquine reduced IL-1β release and diminished interleukin IL-1β mRNA expression in THP-1 cells. Chloroquine also decreased IL-6 and IL-1 in monocytes and macrophages [50]. In addition, chloroquine-induced inhibition of TNF-α production by immune cells happens through disruption of cellular iron metabolism. The drug also suppresses IL-6, IL-12, TNF-α, IFN-α, β and γ and may interfere with the ACE2 the glycosylation of the receptor, thus suppressing the binding of the SARS-CoV-2 to target cells [51].

Clinical studies have recommended a dose of 500 mg of chloroquine twice per day for 10 days, in patients with mild, moderate or severe COVID-19 pneumonia, when there were no contraindications to the drug. In adults 600 mg of chloroquine can be given and then 300 mg after 12 h on day 1. Afterwards, 300 mg/2 days on days 2–5 days and finishing the treatment at day 5 to decrease the risk of side effects, the drug has a long half-life (30 h) [52]. Chloroquine was efficient against SARS-CoV-2 in Vero E6 cells using a 50% and 90% effective concentration (EC50 and EC90 values) of 1.13 μM and 6.90 μM. Therefore, chloroquine improves the evolution of pneumonia in COVID-19 patients [53].

Chloroquine also diminishes the activity of the enzyme quinone reductase 2, which is a structural neighbor of UDP-N-acetylglucosamine 2-epimerase [54], that participates in the sialic acid biosynthesis. This acid is an acidic monosaccharide located at the end of sugar chains that are located on cell transmembrane proteins that participate in ligand recognition. When chloroquine interferes with the biosynthesis of sialic acid could be responsible for the broad antiviral spectrum of that drug since some viruses use sialic acid residues as receptors such as the orthomyxoviruses and the human coronavirus HCoV-O43 [55].

Hydroxychloroquine is different from chloroquine since it has hydroxyl group at the end of the side chain. This molecule can be ingested orally as hydroxychloroquine sulfate. The pharmacokinetics of hydroxychloroquine resembles that of chloroquine, having a rapid gastrointestinal absorption and elimination by the kidney. Hydroxychloroquine has a good tolerance and can be used in high doses for long periods [56].

In the present global infection of SARS-CoV-2, treatment with hydroxychloroquine combined with azithromycin, a macrolide antibiotic, resulted in 100% of patients that were virologically cured in contrast with only 57.1% of subjects treated with hydroxychloroquine alone and 12.5% with the control group. It was therefore concluded that chloroquine and hydroxychloroquine should be used in patients suffering from pneumonia caused by SARS-CoV-2.

Chloroquine has also been proposed as a preventative treatment for COVID-19 [57]. In that study the authors gave 600 mg of chloroquine for 14 days and found a 50% efficacy of hydroxychloroquine, by decreasing the viral load at day 7. They also reported that hydroxychloroquine failed in two patients (mother and son) and concluded that the failure was probably due to a resistance mechanism. However, they did not mention if there were side effects of the drug administration [58].

## 4. Drawbacks of Chloroquine and Hydroxychloroquine

Cardiovascular side effects, particularly prolongation of the QT interval in the electrocardiogram and hypotension have been found with the use of several quinolines and other structurally related antimalarial drugs. A prolongation of the QT interval is a non-specific biomarker of risk for the development of Torsade de Pointe which may turn into potentially lethal polymorphic ventricular tachyarrhythmias. Therefore, there is a need to review the cardiovascular safety of quinoline and other structurally related antimalarials when used massively during a pandemic.

There are currently several clinical trials registered on the effects of the use of chloroquine and hydroxychloroquine. There is a Clinical Trial in which these drugs will be used in combination with others. However, even if it is true that we are in an emergency to combat COVID-19, we must always keep in mind that the cause of death may be due to other factors that are associated with treatment in addition to the infection by the virus. Therefore, doses and interactions of medications must be carefully analyzed independently of their benefic effect on the mechanisms of action against the virus.

The increasing employment of structurally related antimalarials to quinoline in massive treatments to eliminate viruses renders it necessary to study in detail their cardiovascular safety properties and to monitor and analyze the ECG during the treatment [58]. This is further emphasized by the reported studies in patients with lupus erythematosus, where a relation between structural ECG abnormalities and cumulative antimalarial dose above the median (1207 g) were found [59].

Myocardial inflammation and fibrosis could be related to rhythm disorders having different underlying pathophysiology. Cardiomyocyte necrosis can be caused by inflammatory processes and oxidative stress that lead to structural remodeling. Moreover, chronic inflammation links autoimmune disease to autonomic dysfunction that includes sympathetic over-activation and a decrease in parasympathetic function. Cardiac arrhythmia may also be caused by autoantibody-mediated inhibition of type 2 muscarinic cholinergic receptors, β1-adrenergic receptor signaling potassium or L-type calcium currents. In autoimmune disease patients, drug-induced arrhythmias, caused by chloroquine, methotrexate and corticosteroids may also appear [60]. Therefore, the use of this type of drugs is justified in emergency situations; however, medical and interaction surveillance is mandatory since it could lead to mortality due to the treatment.

### Antiviral Drugs

Remdesivir, a nucleotide analog, inhibits the EBOV RNA-dependent RNA-polymerase. Preliminary findings in a total of 499 patients that received treatment with Remdesivir against the disease of EBOV showed a mortality rate of 33% in patients in the early infectious stages. The same authors noted a 75% mortality rate in 1900 infected subjects that were not treated during the epidemic [61].

Remdesivir acts on viral RNA polymerase, evading proofreading by viral exo-nuclease, diminishing viral RNA production. A slowed suspension of the nascent chain of RNA from the virus has been suggested as the antiviral mechanism of Remdesivir. Remdesivir had antiviral activity against many EBOV variants [62]. In-vitro studies showed that Remdesivir suppresses coronaviral replication including that of MERS-CoV and SARS-CoV. Remdesivir was efficient against Bat-CoVs and other circulating human-CoV in human lung cells when it was tested in-vitro using epithelial cell cultures of primary human airways. Another study showed that Remdesivir and INF-β were more effective than lopinavir, ritonavir and INF-β in-vitro and in a MERS-CoV mouse model. Therefore, Remdesivir might constitute an alternative for the therapy of patients with COVID-19 [63]. At present there is an ongoing phase 3 randomized, placebo-controlled, double-blind, multicenter study which is determining the efficiency and safety of Remdesivir in 452 hospitalized adult subjects with severe COVID-19 [64]. In the treatment of SARS-CoV-2, antivirals which include oseltamivir, lopinavir/ritonavir, ganciclovir and ribavirin have been employed [65].

## 5. Anti-Inflammatory Drugs

The release of IL-1 and IL-6 are suppressed by Tocilizumab. Therefore, this drug has a therapeutic effect in many inflammatory diseases, including viral infections [15] and may help in COVID-19. The inflammatory state induced by the SARS-CoV-2 might also be decreased by the overexpression of cytokine IL-37 which abolishes acquired and innate immune response. IL-37 also inhibits inflammation by its action on the IL-18Rα receptor. IL-37 acts on mTOR thus elevating the adenosine monophosphate kinase that has immunosuppressive activity. By inhibiting IL-6, IL-1β, CCL-2 and TNF-α, this cytokine suppress inflammation and the molecules of the class II histocompatibility complex. Another inhibitory cytokine that may help in patients with COVID-19 is the newest cytokine of the IL-1 family, IL-38, which is produced by some immune cells including macrophages and B cells, IL-38 inhibits IL-1β and other pro-inflammatory cytokines [15].

Other drugs that have been proposed against SARS-CoV-2 are JAK inhibitors that have been approved for myelofibrosis and rheumatoid arthritis such as ruxolitinib, fedratinib and baricitinib. These drugs are potent anti-inflammatories that may be useful against the effects of the elevated cytokine levels (including IFN-γ) which are commonly present in people with COVID-19. They act by inhibiting the JAK-STAT signaling and even if the three of them have similar effects, the high affinity for AAK-1of baricitinib suggests that it is the best of the group. This is even potentiated since it can be orally given once a day having little side-effects. Furthermore, baricitinib may be combined with antivirals that act directly (lopinavir or ritonavir and Remdesivir) and which are now being tested in the COVID-19 epidemic, due to its reduced interaction with drugs that metabolize the CYP enzymes. Combinations of baricitinib with direct-acting antivirals could diminish viral replication, viral infectivity and the anormal host inflammatory response [66].

However, these drugs are metabolized in the liver and most of the metabolites produced from chloroquine and hydroxychloroquine sulfate, oseltamivir, ribavirin and lopinavir/ritonavir, are eliminated in urine. Therefore, their metabolism may be impaired by damage to the liver and kidneys [66].

## 6. Corticosteroids

The use of corticosteroids was widespread during the outbreaks of severe MERS-CoV and SARS-CoV-1 and they are now being employed in patients with SARS-CoV-2 in combination with other therapeutics. Combination treatment of low-dose systematic corticosteroids and antivirals and atomized inhalation of interferon are proposed as management of critical patients with COVID-19 [67]. Corticosteroids inhibit lung inflammation and diminish the immune responses while increasing the clearance of the pathogen. Systemic inflammation is associated to adverse outcomes in severe cases of influenza and SARS-CoV infection. In SARS, there is still inflammation after viral clearance. In SARS and MERS infections the pulmonary histology shows that there is inflammation, diffuse alveolar damage and septic shock. These conditions have been reported in patients with COVID-19.

Corticosteroids are commonly employed in septic shock even if there exists uncertainty about their efficiency. Most patients in septic shock trials also show bacterial infection that results in myocardial insufficiency and vasoplegic shock. However, invasive ventilation results in elevated intrathoracic pressure, which results in shock in severe hypoxemic respiratory failure and impairs the filling of the heart and not vasoplegia. Therefore, the treatment using corticosteroid should not be employed for lung injury induced by SARS-CoV-2 [68].

## 7. Arbidol

Arbidol has been used for many years in Russia and is also known as umifenovir. It was approved to treat the influenza viral infections in Russia and China. It has little side effects and is approved for SARS treatment [69]. Arbidol or ethyl-6-bromo-4-I(dimethylamino)methyl-hydroxy1-methyl-2((phenylthio)methyl)-indoe-3 carboxylate hydrochloride monohydrate, is a small indol derivative. It stimulates the immune system increasing serum IFN and activating phagocytes and it also has a direct effect on respiratory viruses.

The arbidol molecule has a broad-spectrum antiviral activity and it inhibits the critical steps of interactions between the host cell and the virus. It acts directly on the virus modifying various stages of the viral life cycle, including cell binding, internalization and replication. Arbidol unites to lipid and protein residues suppressing the entrance of the virus entry and its fusion, budding, replication, assembly and viral resistance [70].

It was invented 40 years ago by a joint consortium of Russian scientists arbidol has been employed in China since 2006 for the treatment and prophylaxis of human pulmonary diseases caused by influenza A and B viruses and of other human pathogenic respiratory viruses and it has been present in the Russian market for 20 years [71]. It has also proven useful for the prevention of the epidemic of poultry flu in China. Its clinical efficacy was first reported and published in Russia in the 1990s [71,72]. It is cost-effective in prophylactic and curative treatments, against acute respiratory viral infections [72].

Arbidol has only been used with a pharmacological approach and therefore, there are insufficient data on its toxicity to rigorously evaluate its safety in chronic administration. Nevertheless, there are many studies that comment its good tolerability. It is not expensive for third world countries that require effective antiviral therapies. In this pandemic situation there is a clinical trial in progress [72].

The combination of antivirals has been tested with other medications such as arbidol and carrimycin and its efficacy against the novel coronavirus infection has been informed. Therefore, its use has been widely proposed ([73] ClinicalTrials.gov. Identifier NCT04275388). Currently several trials compare lopinavir-ritonavir treatment and other drugs for COVID-19—versus ARB ([74] ClinicalTrials.gov, Identifier NCT04295551).

## 8. Danoprevir

Danoprevir was approved in China as an HCV NS3 protease inhibitor. It may be used to treat non-cirrhotic genotype 1b chronic hepatitis C, combined with ribavirin, ritonavir and peginterferon-α [75]. Currently, a trial study has been initiated for its use against COVID-19.

## 9. Importance of Choosing Therapies According to Comorbidities

Patients infected with COVID-19 have a higher risk of dying when there are comorbidities such as diabetes, hypertension, obesity and other pathologies that comprised the MS. Furthermore, there is a higher mortality rate when there is cardiac injury associated to a viral infection as reported in previous studies ([76] ClinicalTrials.gov, Identifier NCT04291729). Therefore, even in an emergency, the mechanisms of action of each drug of the combinatorial therapy should be taken into account. The interactions and doses should also be considered for toxicity since, a greater adverse effect, if not born in mind, could be expected, which could be more dangerous than the disease itself.

Lopinavir/ritonavir can have adverse effects such as increasing the concentration of cholesterol and triglycerides, in the case of subjects with diabetes or elevating the blood sugar concentration. In addition, as has been seen after treatment with an HIV treatment, conditions may change in the immune system as part of the inflammatory syndrome of immune reconstitution which sometimes happens when the immune system begins to recover [77]. The combination of drugs may also induce a lipodystrophy syndrome and increase bleeding episodes in people with hemophilia.

These combinations of drugs should not be administered in conjunction with medications whose clearance depends on CYP3A and for whom the plasmatic concentrations have been associated with severe effects that may become a threat to the life of the patients.

It has also been reported that the COVID-19 infection leads to a higher mortality rate in elderly patients. Nevertheless, this elevated mortality could be associated with comorbidities and to the fact that these patients are usually receiving other medications to treat other conditions and that may be using ritonavir. Some of these conditions might be prostatic growth. Patients with prostatic growth receive treatment with alfuzosine, which, when combined with ritonavir, predisposes to severe hypotension [78]. Ritonavir increases the concentration of alfuzosine and may constitute a confusion factor leading to death. Many elderly patients are also being treated for arrhythmias [79]. The use of amiodarone increases its plasmatic concentrations and increases the risk of arrhythmias or other adverse reactions of drugs used for the treatment of COVID-19. The use of antipsychotics and neuroleptics is also common in elderly patients. Their use together with ritonavir and pimozide increases the risk of severe hematologic diseases. The use of ritonavir with antihistaminic drugs such as astemizol, terfenadine or with drugs that control the gastric motility increases their concentrations and leads to arrhythmias [77].

Furthermore, the use of sedatives such as sildenafil which is also used to control pulmonary arterial hypertension may lead to syncope when combined with ritonavir. Moreover, the concentration and clinical effects of lopinavir and ritonavir may be decreased in patients that use alternative medicine employing compounds that contain the Saint John herb (Hypericum perforatum) [64,78]. Table 1 summarize the effects of the drugs addressed in the manuscript.

## 10. Proposal of Adjuvant Antioxidant Therapy by Our Group

Antioxidant therapy in septic shock was proposed long ago by Hippocrates, who used myrrh (Commiphora mukul, Commiphora myrrha) for therapeutic medicinal and anti-inflammatory purposes [80]. Currently antioxidant therapies are used for various conditions worldwide [81].

In a study by our group, still in the process of being published, we saw that the use of antioxidants showed a statistically significant improvement in patients with pulmonary sepsis. Oxidative stress is elevated during critical illness and the oxidative damage participates in organ failure. The administration of glutamine or antioxidant vitamins, given in the nutritional support or as separate medications during prophylaxis may attenuate OS [81]. In septic shock there is an increase in cardiac output with a decrease in systemic vascular resistance caused by arterial dilation [82]. Excessive or prolonged decrease in vascular resistance can produce hypotension resistant to the use of vasopressors, which can contribute to severe heart failure [83]. Sepsis is an organ dysfunction that threatens life which results from an inadequate response of the host to infection. It consists cellular and metabolic dysfunctions that determine a high mortality [84]. Sepsis is the greatest cause of mortality in ICU worldwide since it reaches up to 80% in patients with multiple organ failure (MOF) [85]. Therefore, in view of our preliminary results and the evidence above mentioned we propose the use of antioxidant therapy as a complementary measure to the pharmacological treatment of COVID-19 where sepsis is present.

Although anti-oxidant therapy has not been tested in COVID-19, the consequences of antioxidant therapy in respiratory failure, particularly in ALI or ARDS have been reported in a previous meta-analysis [86] and it has been concluded that it could help the supportive strategies and lung-protective ventilation which are the only approaches that improve outcomes in patients with COVID-19. Blocking individual proinflammatory cytokines with antibodies or the use of antioxidants independently have not proven to be very useful because of the complex nature of these disease and they only work as a support to ventilation and together with other pharmacological strategies [87].

Nowadays, there is interest in therapies that employ natural compounds which combine antioxidative and anti-inflammatory agents for systemic conditions. Many of the proposed natural agents have pleiotropic effects, such as activating antioxidant defense mechanisms while inhibiting proinflammatory signaling. Moreover, these natural products have been employed in the Asian subcontinent for centuries with little toxicity. However, the pharmacokinetics and pharmacodynamics of these compounds must be tested in human trials before they are implemented as therapies [88]. We propose the use of one or more of the following antioxidant compounds for the treatment of COVID-19.

## 11. *N*-acetylcysteine

Preclinical studies have proposed the administration of *N*-acetylcysteine (NAC), a precursor of glutathione, as a strategy to limit oxidative stress lung injury since it increases the content of intracellular glutathione [89]. A central feature of many lung diseases is the alterations of the metabolism of glutathione that occurs both in the alveoli and lung tissue [90]. NAC increases the synthesis of glutathione synthesis, elevates glutathione-S-transferase activity and has a direct action on free radicals. However, treatment with NAC does not decrease lung tissue myeloperoxidase activity or 3-nitrotyrosine levels [91]. The application of *N*-acetylcysteine decreases the levels of IL-8, IL-6, ICAM and the soluble α receptor for tumor necrosis p55. These mechanisms could be attributed to the adequate control of the inflammatory immune response. In relation to clinical behavior with the use of NAC in patients with septic shock, a shorter mechanical ventilation time and fewer days of ICU stay have been found [92].

The use of liposomes (L-NAC) elevates NAC absorption and intracellular concentration. There were high non-thiol proteins with a pre-treatment with L-NAC (25 mg/kg intravenous) and NAC levels were elevated in homogenates of the lung. Lipid peroxidation, chloramine concentration, damage to ACE, lung edema and the concentrations of thromboxane and leukotriene B2 and B4 were diminished in the lungs after NAC supplementation in LPS-exposed animals [93]. Further studies are needed to establish the most effective way to apply pulmonary antioxidant interventions [94].

In clinical trials, supplementation with a bolus of 150 mg/kg NAC followed by 50 mg/kg/day of NAC for 4 days in patients with acute lung injury ALI or ARDS improved the oxygenation rate from the first to the fourth day and reduced mortality. However, it had no effect on mechanical ventilation time [95]. In a randomized, double-blind, placebo-controlled clinical trial in patients with septic shock, there was improvement in hemodynamic variables and resolution of organ failure in the group of patients who received management with NAC [96]. The hemodynamic effect on biomarkers of inflammation in patients with septic shock and NAC treatment in a double-blind, placebo-controlled clinical trials have been evaluated. The results showed that NAC treatment improved oxygenation, static lung compliance, decreased levels of IL-8 and soluble TNF-α-receptor p55. A shorter mechanical ventilation time and shorter stay in the ICU was found in patients treated with NAC [97].

However, in another randomized, double-blind, placebo-controlled clinical trial conducted in ventilated patients with two or more organ failures. Furthermore, these authors reported an increase in mortality in patients that received treatment after 24 h of hospital admission [98].

The NAC effect on hepatic blood flow, hepatosplenic oxygen transport and liver function in patients with early-stage septic shock was evaluated in another randomized, double-blind, placebo-controlled clinical trial that showed the authors found that hepatosplenic flow and liver function improved after the NAC infusion and this was secondary to the increase in the cardiac index [99]. Another study in patients with sepsis, reported a decreased activation of NFkB associated with decreased levels of IL-8, levels of IL-6 but ICAM-1 showed no significant difference [100]. Another study evaluated the levels of microalbuminuria and organic dysfunction in patients with severe sepsis after the administration of NAC 50 mg/kg/4 h followed by 100 mg/kg/24 h. Microalbuminuria levels were similar in both groups and the NAC group showed an increase in the SOFA score, particularly cardiovascular failure [101].

In another study, there was a reduction of the increased cardiac index and the duration of acute lung injury with the intravenous supplementation (every 8 h/10 days) with 70 mg/kg NAC or 62 mg/kg 2-oxothiazolidine-4-carboxylate, a prodrug of cysteine (OTZ-procysteine). However, mortality was not modified by NAC or OTZ supplementation. Duration of acute lung injury might be shortened by this type of therapy [101,102].

Moreover, the correlations between mortality rate, length of hospital stay, oxygenation ratio and mechanical ventilation time and the antioxidant supplementation may be altered by feeding-tube or tracheal-tube extubations and by a reduced consciousness level. The evaluation of populations above 43 years old and predominantly males in this study also constitutes a limitation.

## 12. Vitamin C and E

Vitamin C is a cofactor for multiple enzymes that is water-soluble [103]. It is absorbed at the intestinal level through the sodium-dependent transporter of vitamin C, filtered freely in the glomerulus and reabsorbed at the proximal tubule level through the same transporter [103]. Vitamin C inhibits the production of OONO^−^ and O_2_^−^ by inhibiting the NAPH oxidase that produces both O_2_^−^ and iNOS mRNA expression. Therefore, it prevents the abundant NO production that generates OONO^−^ in the presence of O_2_^−^ It also decreases pathological vasoconstriction, as well as the loss of vascular permeability by inhibiting the oxidation of tetrahydrobiopterin, which is the cofactor for eNOS, preventing decoupling between NO and eNOS, which generates O_2_^−^ [43]. Another of its effects is to protect against capillary leakage by inhibiting the activation of protein phosphatase 2A, which dephosphorylates occludin which is crucial for the maintenance of tight junctions. Vitamin C has a protective role against mitochondrial permeability, by activating apoptosis pathways. It also inhibits TNF-α and induces ICAM expression, which increases the adhesiveness of leukocytes in the microcirculation [104].

Decreased plasma levels of vitamin C may be due to inadequate intake, acute or chronic consumption secondary to increased oxidative stress or increased loss [105]. Low levels of vitamin C have been documented in critically ill patients in several studies [106]. Furthermore, levels <10 µmol/L have been reported in these patients despite the administration of the recommended daily requirements [103]. Diminished levels of this vitamin are linked to severity of organ failure and mortality [107]. In a study where the safety of vitamin C administration in patients with severe sepsis was evaluated and two different doses of vitamin C were compared (50 mg/kg/24 h and 200 mg/kg/24 h) with placebo. The results showed decreased levels in all septic patients (17.9 ± 24 µM). Plasma levels increased dramatically in both treatment groups. No adverse effects were reported in the patients who received vitamin C infusion. The patients who received vitamin C had a greater decrease in the SOFA score compared to the placebo group. Decreased C-reactive protein and procalcitonin were reported in the vitamin C group. Lower levels of thrombomodulin were found in the vitamin C group compared to placebo. 28-day mortality was also lower in the low-dose vitamin C group (38.1%) compared to the high-dose group (50.6%) and the placebo group (65.1%). There was no difference in terms of mechanical ventilation days or days of ICU stay. A 96-h infusion of vitamin C in contrast to placebo did not improve vascular injury, organ dysfunction scores and inflammation markers in a preliminary study of patients with sepsis and ARDS. The authors concluded that further research was necessary to evaluate the potential role of vitamin C for other outcomes in ARDS [108].

However, diets enriched with antioxidants (320 IU/L vitamin E, 840 mg/L vitamin C and 320 mg/L taurine), eicosapentaenoic acid (4.5 g/L) and gamma-linolenic acid (4.3 g/L) showed a lower relative risk of death. Mechanical ventilation time and length of intensive care ICU stay were also reduced by this type of diet. This diet also decreased the number of dysfunctional organs [109].

A diminished relative risk of pulmonary morbidity and nosocomial pneumonia was found in patients receiving 60 IU/L α-tocopherol (vitamin E) and 340 mg/L vitamin C through an oro-gastric tube. There was a lower multiple organ failure. Vitamin E and vitamin C administration also decreased the length of ICU stay. However, no significant alterations in isoprostane concentration, white blood cell count, tumor necrosis factor concentration, IL-1 concentration and IL-6 concentration were found, despite a clear tendency for them to be reduced. This supplementation also increased IL-8 concentration [110]. Vitamin E is the most important lipophilic antioxidant in the cell membranes and it prevents lipid peroxidation [111]. Vitamin C regenerates vitamin E increasing its antioxidant effect, avoiding lipid peroxidation and sequestering fat-soluble ROS. In a double-blind, placebo-controlled clinical trial [112], the authors found a significant decrease in mortality at 28 days and fewer days of mechanical ventilation in patients treated with vitamin C and vitamin E. Another randomized, double-blind, placebo-controlled study found a significant decrease in days of mechanical ventilation in patients who were treated with vitamin C and vitamin E [113]. The combined effect of therapy with vitamin E and vitamin C in severe post-operative patients resulted in a decrease in the incidence of ARDS or pneumonia, a decrease in organ failure and a tendency to decrease mortality at 28 days. The most relevant vitamin E function is its antioxidant effect. However, it has other important effects such as stabilizing the cell membrane and maintaining an adequate immune response to infection [114]. We believe that the combined treatment with vitamin E and C is a very promising option that can further decrease the organic damage caused by oxidative stress and systemic inflammation in the lung which results from COVID-19 [115].

## 13. Melatonin

*N*-acetyl-5-methoxytryptamine also called melatonin (MT) is the main endogenous product of secretion of the pineal gland in vertebrates, which is released into the circulation and cerebrospinal fluid [116]. MT has been proposed as a sepsis control agent. Its wide extracellular and intracellular distribution and its synthesis in various of organs can explain the MT role in modulating in numerous physiological processes through various mechanisms. To date, we are not aware of studies in humans with septic shock using MT [117]. MT has been shown to possess ROS sequestration properties, which protect lipids of cell membranes, cytosol proteins and nuclear and mitochondrial DNA [116,117]. In vitro and in vivo studies have shown that it can prevent lipid peroxidation [118], preserve the permeability of the membrane by increasing its fluidity [119] and reduce levels of hydroperoxide in mitochondria by restoring glutathione homeostasis and mitochondrial function in organelles under oxidative stress [120]. Also, MT is also able to stimulate gamma-glutamyl cysteine synthase; therefore, it can increase the intracellular glutathione synthesis [121]. Another of its antioxidant mechanisms is through the restoration of activity of mitochondria that is diminished in some diseases, reducing the consumption of O_2_ by liver mitochondria. Thus, it protects organs from excessive oxidative damage [122].

Regarding acute lung injury secondary to sepsis, an increase in ROS and RNS has been described, derived from the increase in the decoupling of eNOS and iNOS overexpression [123], which is associated with an increase and release of proinflammatory mediators such as TNF-α, IL-6 and IL-9 by bronchial epithelial cells. This leads to macrophages [124], activating neutrophils and other macrophages which cause alveolar destruction and collapse of small airways. These changes can induce endothelial damage, increased pulmonary capillary permeability and pulmonary edema causing deterioration in gas exchange [125].

## 14. Quercetin

2-(3,4-dihydroxyphenyl)-3,5,7-trihydroxy-4H-chromen-4-no, also called, quercetin (QRC) is a plant derivative belonging to the group of polyphenols called flavonoids. It is an aglycone. The main natural sources of QRC include onion, apple and black tea. It has antiviral, cardioprotective, antioxidant, anti-inflammatory and anti-carcinogenic effects [126]. In the small intestine QRC is absorbed and is immediately metabolized by enzymes in epithelial cells and subsequently by the liver. The catechol group of the QRC is methylated at position 3′–4′ by catechol-O-methyl transferase resulting in the formation of isorhamnetin (3’OCH3–QRC) or tamarixetine (4′OCH3–QRC). These metabolites together with the QRC can be conjugated in their hydroxyl groups with glucuronic acid or sulfate by the UCP-glucuronosyltransferase or sulfotransferase, respectively [127]. The half-life of plasma QRC in humans is between 17 and 18 h. Regarding the distribution of QRC (in a dose of 50 and another of 500 mg/kg for 11 weeks) its highest levels are found in the lungs and the lowest levels in the brain and white adipose tissue in rats, whereas concentrations of the metabolite aglycone isorhamnetin were higher in the liver [128]. QRC is safe for most people. However, in doses of up to 1 g daily, it may induce side effects such as headache and paresthesias in the hands and legs, which are reversible. Safety is only affected with the use of high doses, due to the risk of renal toxicity [129]. Safety is only affected with the use of high doses, due to the risk of renal toxicity. On the other hand, its interaction with antibiotics such as quinolones, with anti-neoplasics and immune-suppressants such as cyclosporine, in which their side effects increase, is known. There is also interaction with the use of diclofenac, celecoxib, fluvastatin, glipizide, ibuprofen, irbesartan, losartan phenytoin and others with cytochrome p450 2C9 substrate (CYP2C9) and with p450 2C (CYP2C8) such as paclitaxel, rosiglitazone, amiodarone and verapamil. However, QRC is currently a promise as a flavonoid with multifaceted therapeutic application that can be used in this pandemic from the initial asymptomatic stage, since it can be obtained through the diet, with foods that include grapes, strawberries, blackberries, the of fennel, lettuce and beans [130]. The QRC, present in H. cordata, inhibits several viruses including herpes simplex, mengovirus, pseudorarabias, parainfluenza type 3, Sindbis virus and respiratory syncytia [131]. QRC is able to inhibit the H^+^-ATPase of the lysosomal membrane and, therefore, prevent the removal of the virus [127]. Furthermore, QRC inhibits the ATPase of proteins related to resistance to many drugs elevating the bioavailability drugs against cancer and viruses in vivo [127]. Therefore, QRC may be considered is potentially effective as an antiviral [131].

## 15. Pentoxifylline

Pentoxifylline is a xanthine indicated in some severe cases of alcoholic hepatitis. It also acts on the plasma membrane of red blood cells rendering it more malleable, thus improving blood perfusion. Pentoxifylline exerts several antioxidant and anti-inflammatory activities; it restores glutathione levels, it maintains mitochondrial viability, it inhibits TNF-α production, it preserves vascular endothelial functions. It also improves oxygenation rates and induces a stronger immune response [132]. In patients with ALI/ARDS it diminishes the length of hospital stay, the length of ICU stays, the rate of multiple organ dysfunctions, the mechanical ventilation time and the mortality rate [133]. In a randomized controlled study that included 120 newborns with a mean gestational age of 30 weeks, administration of pentoxifylline (5 mg/kg/h iv for 6 h for 6 days) diminished levels of CRP and TNF-α, decreased duration of respiratory support and antibiotic treatment, diminished the need for vasopressors, resulted in a shorter hospitalization time and decreased incidence of metabolic acidosis, thrombocytopenia and disseminated intravascular coagulopathy [134]. However, no difference in short-term morbidity was observed between treated with pentoxifylline and untreated neonates with sepsis [135]. In a meta-analysis involving 6 studies, with the pentoxifylline administration of in septic infants, pentoxifylline was shown to be effective in decreasing length of hospital stay and reducing all-causes of mortality. There was also a significantly diminished mortality in preterm infants, neonates and infants with proven sepsis and in infants with gram-negative sepsis in a subgroup analysis that resulted in the conclusion that pentoxifylline may be a beneficial adjuvant treatment in sepsis [136] and could also be used in COVID-19.

Treatment with pentoxifylline is contraindicated in patients with recent cerebral or retinal hemorrhage or with active hemorrhage, in patients with coronary artery disease and in patients with impaired renal or hepatic function. Therefore, concomitant use of pentoxifylline, with agents that interfere with platelet function and with coagulation factors such as coumarin, heparin, indandione derivatives, cefotetan, cefamandole, valproic acid and plicamycin may increase the risk of bleeding. Combined use with methylxanthines and sympathomimetic agents can lead to overstimulation of the central nervous system [133]. Table 2 summarizes the effects of the antioxidants on the patients by SARS-Cov-2 infection addressed in the manuscript.

### Proposal of Therapeutic Management with Antioxidants

The design of the study testing the usefulness of antioxidants as an adjuvant in the treatment of COVID-19 must be non-randomized regarding the use of each of the antioxidants, which may or may not be used with pentoxifylline. Its concomitant use may or may not be added to the use of macrolides, hydroxychloroquines or antivirals in the standard dose. The decision must be based on the baseline comorbid condition of the patient infected with SARS-Cov-2 and medications used for chronic diseases at the time of admission. To do this, a therapeutic decision tree and an individualized management algorithm for each patient must be carried out. Table 3 describes the proposal algorithm for therapeutic management with antioxidants for COVID-19 patients.

In summary antioxidant supplementation has been reported to cause better oxygenation rates, increased glutathione and a stronger immune response. There is a reduction of the mechanical ventilation time, length of hospital stay, length of ICU stays, mortality rate and multiple organ dysfunction rates in ALI/ARDS patients. Nevertheless, more studies are still needed to reinforce the benefits of antioxidant supplementation. The use of these complementary measures in the present COVID-19 pandemic has not previously been proposed or used.

## 16. Conclusions

Corona Virus Disease-2019, which results in a severe pneumonia that develops into pulmonary alveolar collapses in a few h and leads to the cessation of oxygen exchange, is caused by Severe Acute Respiratory Syndrome Coronavirus 2. SARS-CoV-2 is a member of the β-coronaviruses. Recognition by a receptor in the host, protease cleavage of the host membrane and fusion determine the transmissibility and pathogenesis of the virus. The ACE2 receptor is the functional receptor for SARS-CoV-2 in all target organs. The mechanism of infection by the virus correlates with the comorbidities that increase the death rate of the infection since the ACE2 receptor number is altered by the diseases or their treatment. Many therapeutic measures have been proposed and are being tested for COVID-19 patients. We suggest the use of antioxidant drugs to improve the conditions of these patients. Oxidizing agents come from the phagocytic leukocytes such as neutrophils, monocytes, macrophages and eosinophils invading the tissue. Free radicals promote cytotoxicity injuring cells. They degrade essential cellular components and alter the protease/antiprotease balance in the tissue interstitium. They also trigger the mechanism of inflammation following an initial insult. In COVID-19, there is an intense inflammatory response that is known as a cytokine storm. Oxidative stress mediates the activation of NFkB inducing in turn the transcription of certain genes thus promoting cytokine production. Release of these cytokines enhances the inflammatory response. Although antioxidant therapy has not been tested in this disease, consequences of this measure in respiratory failure, particularly in ALI/ARDS, have been reported. It improves oxygenation rates; increases glutathione and promotes a stronger immune response. There is a reduction of the mechanical ventilation time, length of hospital stay, length of ICU stays, mortality rate and multiple organ dysfunction rates in ALI/ARDS patients and therefore it could also help patients with severe COVID-19.

## Figures and Tables

**Figure 1 medicina-56-00386-f001:**
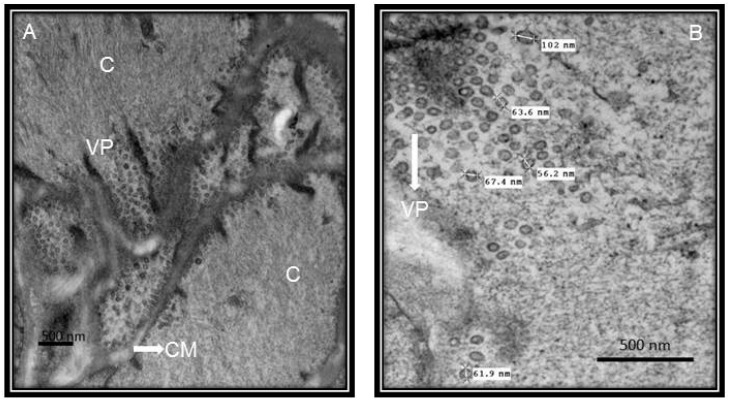
Ultrastructural lung features from a postmortem patient of COVID-19. The patient was a 68-year-old female from our Institute, with type II diabetes, ischemic heart disease, hypertension and morbid obesity as associated comorbidities. The pictures of the electron microscopy panels (**A**,**B**) were made at 25,000× and 50,000× respectively with a Jeol JEM-1011 electronic microscope. Panel (**A**) shows the separation of the two muscular cells of the middle zone of a bronchiole with viral particles. Panel (**B**) shows the middle zone of a bronchiole with viral particles, the viral particles found measured between 56.2 and 102 nm and this agrees with the reported size of the virus. Abbreviations: C = cellule, CM = cellular membrane, VP = viral particle.

**Figure 2 medicina-56-00386-f002:**
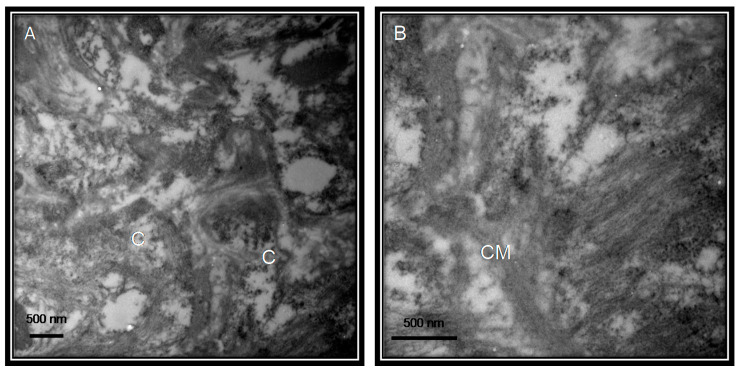
Postmortem lung biopsy of a 28 year old male patient with acute myocardial infarction and aortic stenosis as control subject. The bronchiole without viral particles. The pictures of the electron microscopy panels (**A**,**B**) were made at 25,000× and 50,000× respectively with a Jeol JEM-1011 electronic microscope. Abbreviations: C = cellule, CM = cellular membrane.

**Figure 3 medicina-56-00386-f003:**
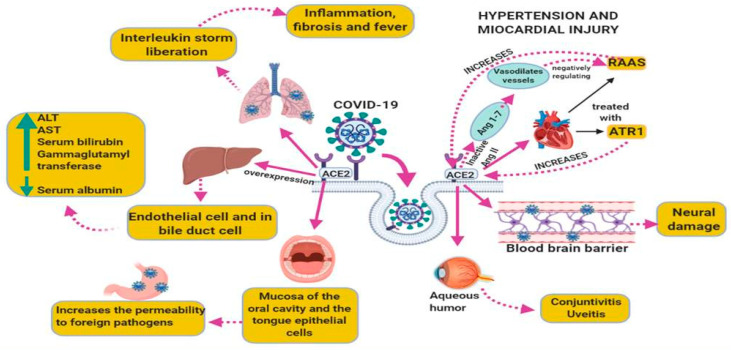
Interaction of SARS-COV-2 with the ACE2 receptor, target organs and its effect on them. Abbreviations: ACE2 = Angiotensin converting enzyme 2, RAAS = Inhibitors of the renin-angiotensin-aldosterone system, ATR1 = receptor type 1 losartan. ALT = alanine aminotransferase, AST = aspartate aminotransferase.

**Figure 4 medicina-56-00386-f004:**
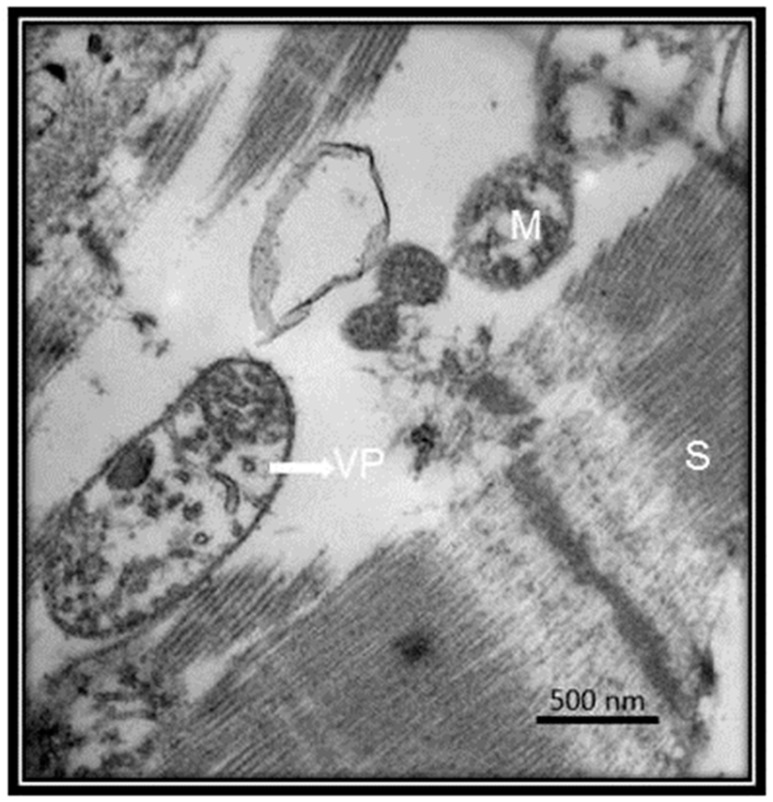
Ultrastructural features of a heart sample from a patient with COVID-19 postmortem. The patient was a 68-year-old female of our Institute that had as associated comorbidities type II diabetes, ischemic heart disease, hypertension, morbid obesity. The picture to 40,000× of the electron microscopy was made with a Jeol JEM-1011 electronic microscope. The Picture describes inter fibrillate mitochondria with viral particles in the heart with loss of cytoplasmic content and loss of internal membrane and loss and shortening of chalk with alterations of the external membrane, rupture of heart fibers. Abbreviations: M = Mitochondria, S = Sarcomere, VP = viral particle.

**Figure 5 medicina-56-00386-f005:**
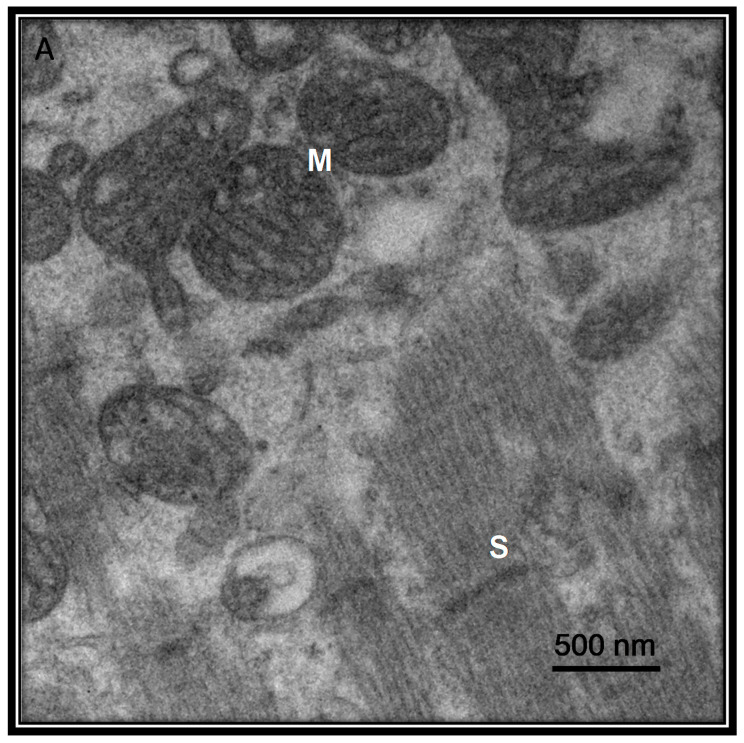
Ultrastructural features of a heart sample from biopsy postmortem of patient 54 years old female with atrial fibrillation as control subject without viral particles. The mitochondria show regular ridges and shorts sarcomere. The picture to 40,000× of the electron microscopy was made with a Jeol JEM-1011 electronic microscope. Abbreviations: M = Mitochondria, S = Sarcomere.

**Figure 6 medicina-56-00386-f006:**
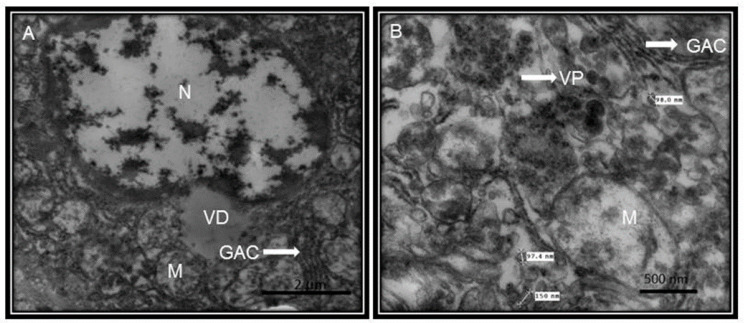
Ultrastructural liver features from postmortem of COVID-19 patient. The patient was a 68-year-old. The pictures of the electron microscopy panels A and B were made at 25,000× and 50,000× respectively with a Jeol JEM-1011 electronic microscope. The panel (**A**) shows vaccination drops from viral damage. The panel (**B**) shows mitochondrial damage, mitochondria with loss of cytoplasmic content, internal membrane and loss of chalk with alterations of the external membrane. The measurement of something viral particles they were found between 98 and 150 nm and agrees with the reported size of the virus. Abbreviations: GAC = Golgi apparatus cisterns, N = nucleus, M = Mitochondria, VD = vaccination drops, VP = viral particle.

**Figure 7 medicina-56-00386-f007:**
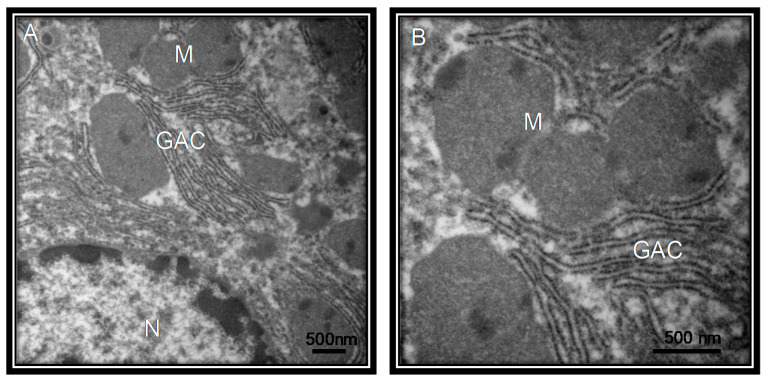
Postmortem lung biopsy of a 28 year old male patient with acute myocardial infarction and aortic stenosis as control subject. The liver without viral particles. The pictures of the electron microscopy panels (**A**,**B**) were made at 25,000× and 50,000× respectively with a Jeol JEM-1011 electronic microscope. Abbreviations: GAC = Golgi apparatus cisterns, N = nucleus, M = Mitochondria.

**Table 1 medicina-56-00386-t001:** Treatment, action mechanism and adverse effect of the various drugs addressed in this review.

Treatment		Mechanism of Action	Adverse Effect
Chloroquine and Hydroxychloroquine	↓	IL-1β mRNA expression in THP-1 cells.	Cardiovascular effects, particularly prolongation of the QT interval in the electrocardiogram and hypotension have been found with the use of several quinolines and other structurally related antimalarial drugs.
		IL-6 and IL-1 in monocytes and macrophages and decreases the activity of the enzyme quinone reductase 2 [50].	
	Ѳ	The replication of some DNA and RNA viruses [52,53].	
		Suppresses the entrance of viruses into host cells and inhibits glycosylation of viral proteins [48].	
		Alkalinizing the endosomal pH in the phagolysosome and by blocking the glycosylation of cellular receptor of SARS-CoV.	
		Suppresses phosphorylation of caspase-1 and of the p38 mitogen-activated protein kinase in THP-1 cells.	
		Suppresses IL-6, IL-12, TNF-α and IFN-α, β and γ [50,51].	
		Suppresses the binding of the SARS-CoV-2 to target cells [51].	
Remdesivir	Ѳ	Suppresses coronaviral replication including that of MERS-CoV and SARS-CoV [64].	Increase in plasma glucose levels.
		Remdesivir acts on viral RNA polymerase evading proofreading by viral exo-nuclease, diminishing viral RNA production [64,65].	
Anti-Inflammatory Drugs	↓	Combinations of baricitinib with direct-acting antivirals could diminish viral replication, viral infectivity and the anormal host inflammatory response [66].	These drugs are metabolized in the liver and most of the metabolites produced are eliminated in urine. Therefore, their metabolism may be impaired by damage to the liver and kidneys.
(Ruxolitinib, Fedratinib and Baricitinib)	Ѳ	Suppresses the JAK-STAT signaling [66].	
Corticosteroids	↑	Decreases the immune responses while increasing the clearance of the pathogen [67,68,69].	Systemic inflammation is associated to adverse outcomes in severe cases.
	Ѳ	The lung inflammation [67,68,69].	The treatment using corticosteroid should not be employed for lung injury induced by SARS-CoV-2.
Arbidol/Umifenovir	Ѳ	Arbidol unites to lipid and protein residues suppressing the entrance of the virus entry and its fusion, budding, replication, assembly and viral resistance [70,71,72].	Arbidol has only been used with a pharmacological approach and therefore, there are insufficient data on its toxicity.
Danoprevir	Ѳ	Is an HCV NS3 protease inhibitor [73].	

↑ Increases; ↓ Decreases; Ѳ Suppresses.

**Table 2 medicina-56-00386-t002:** Antioxidant and clinical effect, that is propose as adjuvant therapy for the treatment of patients infected with COVID-19.

Treatment	Antioxidant Effect		Clinical Effect
NAC	↑	The synthesis of glutathione and glutathione-S-transferase activity [100].	Lipid peroxidation, chloramine concentration, damage to ACE, lung edema and the concentrations of thromboxane and leukotriene B2 and B4 were diminished in the lungs.
	↓	The levels of IL-8, IL-6, ICAM and decreases activation of NFκB [101,102].	
Vitamin C and E	↑	The adhesiveness of leukocytes in the microcirculation [108,109].	The combined effect of therapy with vitamin E and vitamin C in severe post-operative patients resulted in a decrease in the incidence of ARDS or pneumonia, a decrease in organ failure and a tendency to decrease mortality at 28 days.
	Ѳ	The production of OONO− and O2− by inhibiting the NAPH oxidase, the activation of protein phosphatase 2A and TNF-α [109,110,111,112].	
MT	↑	The intracellular glutathione synthesis [122].	Melatonin enhances the immune response by improving proliferation and maturation of natural killing cells, T and B lymphocytes, granulocytes and monocytes in both bone marrow and other tissues and also presents anti-inflammatory action and induces the up-regulation of Nrf2 with therapeutic effects on hepato protection and cardio protection.
	↓	The levels of hydroperoxide in mitochondria by restoring glutathione homeostasis and mitochondrial function in organelles under oxidative stress [120,123].	
QRC	Ѳ	The H+ −ATPase of the lysosomal membrane and the ATPase of proteins related to resistance to many drugs elevating the bioavailability drugs [126,127,128].	It has antiviral, cardioprotective, antioxidant, anti-inflammatory and anti-carcinogenic effects.
Pentoxifylline	↑	The glutathione levels, it maintains mitochondrial viability [133].	Decreases duration of respiratory support and antibiotic treatment, diminished the need for vasopressors, resulted in a shorter hospitalization time and decreased incidence of metabolic acidosis, thrombocytopenia and disseminated intravascular coagulopathy.
	↓	The levels of CRP and TNF-α [134].	
	Ѳ	TNF-α production [132].	

Abbreviations: NAC = *N*-acetylcysteine, ACE = Angiotensin converting enzyme, MT = Melatonin QRC = quercetin, ↑ Increases, ↓ Decreases, Ѳ Inhibits.

**Table 3 medicina-56-00386-t003:** The algorithm proposal for therapeutic management with antioxidants for COVID-19 patients.

Identification of the Case and Integral Study	Management by the Intensive Care Specialist	Proposed Treatment Assays	Additional Proposed Antioxidant Therapy to the Medical Management in COVID-19 Patients with Pneumonia	Drug Interactions
1. Identification of infected cases having a medical history (healthy or with simple or combined comorbidities), age, body mass index. 2. Integral laboratory tests, including total cholesterol, HDL, LDL, triglycerides, D dimer, troponins, natriuretic peptide, creatine phosphokinase, Cl, Na, K, magnesium, phosphorus. Evaluation of hepatic and renal function. 3. Chest X-ray and/or computerized tomography. 4. Cardiovascular: electrocardiogram, echocardiogram. 5. When a cardiovascular disease is present, a multidisciplinary management involving a cardiologist is suggested (this applies to all cases).	1. Antibiotics, according to requirements. A culture if an associated bacterial infection is present. 2. Mechanical ventilation. 3. Hemodynamic Management. 4. Extracorporeal membrane oxygenation if necessary (this applies to all cases).	The patient may be treated with any of the therapies suggested by the clinical assays in standard doses. Only one or a combination of two antioxidants may be applied. When the pulmonary function is compromised, a combination of NAC and vitamin C is suggested. (This applies to all cases).	NAC dose: 600 mg/24 h orally or through a nasal enteral tube for 5 days. It should not be used when there is a gastric ulcer or antecedents of asthma or a cysteine allergy.	Anti-cholinergic drugs, anti-histaminic drugs.
MT dose: 5 mg tablet (10 capsules)/24 h orally or through a nasal enteral tube for five days. It is safe and has a low toxicity. It is contraindicated during pregnancy, diabetes mellitus and when a patient has high systemic hypertension of the blood, and care must be taken when it is used together with some antihypertensive drugs.	Sedatives, Luvox contraceptives and cocaine increase the effect of MT. Verapamil, nifedipine, caffeine, alcohol and immunosuppressors decrease the MT effect.
Vit C dose: 1 g every 6 h orally or through a nasal enteral tube for 5 days. It should not be used in patients with gastric ulcers, falciform anemia or during pregnancy. At high doses > 3 g, it may interact with medications used in patients with diabetes mellitus, hypertension, or Parkinson’s disease.	Aluminum, estrogens, protease inhibitors, antivirals: nelfinavir amprenavir, ritonavir, saquinavir, statins, glimepiride, glipizide, metformin, pioglitazone, rosiglitazone, antihypertensive drugs.
Vit E, dose capsules of 400 UI that are the equivalent of 400 mg every 12 h for 5 days. It should not be used when retinitis pigmentosa is present.	Simvastatin and niacin, chemotherapy, anticoagulants and antiplatelet agents and non-steroidal anti-inflammatory drugs.
QRC, at a dose of 500 mg every 12 h. It is safe for most people, but is not contraindicated during pregnancy. The dose should be adjusted in patients with renal failure.	Quinolones, uroseptal, cyclosporine, celecoxib, diclofenac, ibuprofen, Fluvastatin, irbesartan, losartan.
Initial evaluation. 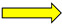	Measures that are jointly evaluated according to the initial evaluation. 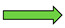	Measures that are jointly evaluated according to the initial evaluation. 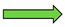	Pentoxifylline (vasodilator, inhibitor of IL-6), at a dose of 400 mg daily orally or IV. Patients that are intolerant to methylxanthines might not tolerate it. (The doses of the elected antioxidants and/or pentoxifylline are similar in all cases. The selection must be taken according to contraindications and drug interactions). Measures that are jointly evaluated according to the initial evaluation. 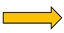	Coumarin, heparin, indandione derivatives, cefotetan, cefamandole, valproic acid and plicamycin may increase the risk of bleeding. Final evaluation. 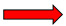
**Evaluate and Apply According to the Patient’s Comorbidities**
**Diabetes Mellitus with or without Hypertension**NAC or MT or QRC or vit E + pentoxifylline.	**Hypertension**NAC or MT (if antihypertensive drugs that may interfere are not being used) orQRC or vit E (if anticoagulants are not being used) or pentoxifylline.	**Obesity****Without Another Comorbidity**NAC or MT or QRC or vit E or vit C + pentoxifylline.**Obesity + SAH + DM**NAC or QRC + pentoxifylline.	**Renal Insufficiency****without Substitute Management****(Dose Adjustment)**NAC or MT or QRC or vit E + pentoxifylline.**In Patients in Dialysis of Hemodialysis**Standard dose.	**Heart Failure**NAC or MT or QRC or vit E + pentoxifylline.Monitor if the patient uses antiplatelet or anticoagulant digoxin or has an arrhythmia.
**Ischemic Heart Disease**NAC or MT or QRC or Vit E + PentoxifyllineAdjust if there is use of antihypertensive drugs that interact with any of them	**Myocarditis**NAC or MT or QRC or Vit E + Pentoxifylline**On Dialysis or Hemodialysis**Standard dose. Immunoglobulin should be the treatment of choice except contraindication	**Arrhythmias**NAC or MT or QRC or + Pentoxifylline	**Chronic Obstructive Pulmonary Disease**NAC or MT or QRC or + Pentoxifylline**Smoking**NAC or MT or QRC or Vit C + Pentoxifylline	**Immune****Suppression**NAC or Vit E (if anticoagulants are not being used) or Pentoxifylline**Cancer**NAC or QRC

Abbreviations: IV = Intravenous, NAC = N-acetylcysteine, MT = melatonin, QRC = quercetin, vit C = vitamin C, vit E = vitamin E, SAH = systemic arterial hypertension, DM = diabetes mellitus, HDL = high-density lipoprotein, LDL = low-density lipoprotein.

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
