# Peer review of "Is Antioxidant Therapy a Useful Complementary Measure for Covid-19 Treatment? An Algorithm for Its Application"

_medicina, 2020, doi:10.3390/medicina56080386_

Round 1

Reviewer 1 Report

In this review, Soto et al., discussed the general characteristics of the virus, its mechanism of action, and the way in which the mechanism correlates with the comorbidities that increase the death rate. They also discuss the currently proposed therapeutic measures and propose the use of antioxidant drugs to help COVID-19 patients. 

SARS-CoV2 and its related disease had changed the way that we produce information in science. Several papers are published daily in regards to this issue. Thus, if the authors intend to do a review, the review needs to have high quality and update references. Also, the review needs to reflect the strength of the researcher group. I started with these words because the whole manuscript must be deeply revised, re-write, and re-formatted to reach the quality level for a publication. 

The manuscript did not reflex the title. In fact, there is no algorithm to possibly use antioxidant therapy in COVID-19 patients (said table 3 and there is not table 3). 

The authors have a strong background in cardiovascular diseases (showed by more than 50 publications). Thus, this should be the focus of this review avoiding getting into a field that will not apport any new knowledge and that for the lack of experience could decrease the quality of the manuscript (Virology: lines 33 to 125). All this information could be summarized in less than 10 lines. Also, there are several reviews regarding these topics. I strongly suggest deleting all this introduction that did not help the manuscript's main idea. 

The introduction and body of the review need to focus on describing whether or not the Antioxidant therapy could have a positive effect on COVID-19. First, the author needs to describe the role of ROS in the pathogenesis of SARS-CoV2. Then, whether antioxidant therapy could revert that. https://doi.org/10.1016/j.arcmed.2020.04.019, DOI: 10.23937/2474-3658/1510121,doi:10.1016/j.toxrep.2020.06.003,DOI: 10.3389/fcell.2020.00479

There are several missing references and other references need to be changed for primary references (you need to go to the primary source of information). 

The figures should include controls to compare or other biochemical markers that validate that the reflected as Viral particles are viral particles. 

All the clinical trials that are mentioned in the work must have the identification number obtained from Clinicaltrials.gov

All the tables need to be re-designed. Add divisions for each drug to follow the message easily and decrease the information burden. Also, add the references in numbers.

The authors need to re-write the abstract making clear to the main objective of the review. The same with the conclusion avoiding repeat information of the previous sections and focusing on the main importance of the work.

Other minor comments are shown in the following:

Line 132: the authors refer to Spike-TMPRSS2 as 'It also binds to a host cellular trans-membrane serine protease''. This statement needs to be changed. TMPRSS2 is not a binding receptor of the viral spike. The viral spike after binding to ACE2 interacts with TMPRSS2 and is cleavage. Also, the citation should be hoffmann et al 2020 https://doi.org/10.1016/j.cell.2020.02.052 

Line 145:The ACE2 receptor-related signaling pathways play an important role in several pathologies that are considered as comorbidities that increase the death rate of COVID-19. There is a missing reference that supports this statement.

Line 146:The fact that the virus binds to this receptor and activates the signaling pathways might explain, in part, why symptoms are more pronounced in patients with CVD and related diseases including MS, diabetes,hypertension and hepatic diseases [15]. First, reference 15 did not reflect this statement. Second, the virus bind to ACE2 but is it related to activation of the intracellular signaling? If that is true, the author needs to show the respective reference that supports that. 

Line 156: COVID-2. Change to COVID-19

Sections COVID-19 and Diabetes/Brain did not apport any new knowledge, interpretation or support the main idea of the manuscript. I suggest either re-structure and adding more detail or delete. 

Line 221:COVID-19 severity is associated with liver diseases such as cirrhosis, non-alcoholic fatty liver disease, alcohol-related liver disease and chronic viral hepatitis[32]. Reference 32 did not reflect the author's statement. The author must re-write their statement or add the appropriated reference 

Line 229: Reference 34 did not reflect the author's statement. The author must re-write their statement or add the appropriated reference 

Line 231: There is also an elevation of gamma-glutamyl transferase which is a diagnostic biomarker for cholangiocyte injury, in 30 patients out of 56 (54%) with COVID-19 that required hospitalization. Reference???

Line 233: Reference 34 did not reflect the author's statement. The author must re-write their statement or add the appropriated reference 

Line 245,260 Reference 36 did not reflect the author's statement. The author must re-write their statement or add the appropriate reference. The presence of a receptor did not means infection. 

Line 264: Reference 37 did not reflect the author's statement. The author must re-write their statement or add the appropriate reference. There is no direct relation to SARS-CoV2 in this reference.  

Line 270: Check the title. There is not an inflammatory system. Also, this section needs to be deeply revised to show the importance related to the main idea of the review.

Line 319 and 335: What is the importance of these sections to the main idea of the work? it should be deleted. 

Line 428-441: This statement did not contribute to gain a deep understanding of HCQ therapy in SARS. It should be deleted. 

The antiviral section needs to be restructured as well. The author mentioned some drugs as teicoplanin and carriomycin that are far to be used in antiviral therapy against SARS-CoV2. There several reviews in regard to clinical trial and SARS-CoV2. The author need to check that first. doi: 10.3390/ijms21072657. doi: 10.1128/AAC.00483-20 doi: 10.1002/phar.2398  doi: 10.21873/invivo.11949. doi: 10.1016/j.lfs.2020.117883 among other... If the authors want to include some information about drugs, should be related to the main candidates to be used in the therapy (actually in ongoing clinical trials). Carriomicin clinical trial started in February and it is not recruiting yet!

The section of drawbacks of combinatorial therapies needs to be carefully revised. This section focuses mainly on the side effects of protease inhibitors (lopinavir/rito). That is not a combinatorial therapy (lopinavir is a protease inhibitor and ritonavir is added to boost the effect of lopi by decreasing the metabolism. Thus, it is considered as a single drug)!!!! The author just described the interaction between other drugs and the antivirals. There are several examples of combinatorial therapy against SARS-CoV2 that are actually in clinical trials (more than 100 clinical trials registered in the NCBI) i.e: HCQ/camostat, lopi/rito/HCQ, Famotidine/HCQ, Arbidol/lopi/rito, Nitazoxanide/lopi etc. 

Line 812: Reference 137 did not reflect the author's statement. the authors need to go to the primary source.  

Line 822:QRC is safe for most people. However, there is no definitive evidence of its use in pregnancy. It is safe in doses of up to 1 gram daily, but it may induce side effects such as headache and paresthesias in the hands and legs, which are reversible. Safety is only affected with the use of high doses, due to the risk of renal toxicity. This statement needs to be re-write.. It is not clear the message. 

I am not a native speaker and I found several grammatical errors and in instances poor redaction. So, the language needs to be review by a native speaker to improve the quality and readability. 

For this reviewer, the main idea of the review is interesting, but the manuscript is far to meet the quality criteria. The manuscript needs to be deeply revised, modified, and improved to be accepted for publication. 

Author Response

Referee 1

In this review, Soto et al., discussed the general characteristics of the virus, its mechanism of action, and the way in which the mechanism correlates with the comorbidities that increase the death rate. They also discuss the currently proposed therapeutic measures and propose the use of antioxidant drugs to help COVID-19 patients. 

Question 1

SARS-CoV2 and its related disease had changed the way that we produce information in science. Several papers are published daily in regards to this issue. Thus, if the authors intend to do a review, the review needs to have high quality and update references. Also, the review needs to reflect the strength of the researcher group. I started with these words because the whole manuscript must be deeply revised, re-write, and re-formatted to reach the quality level for a publication. 

Answer:

R-Thanks for the suggestions, we revised the manuscript and restructured it.

Question 2

The manuscript did not reflex the title. In fact, there is no algorithm to possibly use antioxidant therapy in COVID-19 patients (said table 3 and there is not table 3).

Answer:

R-We apologize, we suspect that when the manuscript was sent, table 3 that showed the algorithm was not properly loaded. In this new version we have made sure it is included.

Question 3

The authors have a strong background in cardiovascular diseases (showed by more than 50 publications). Thus, this should be the focus of this review avoiding getting into a field that will not apport any new knowledge and that for the lack of experience could decrease the quality of the manuscript (Virology: lines 33 to 125). All this information could be summarized in less than 10 lines. Also, there are several reviews regarding these topics. I strongly suggest deleting all this introduction that did not help the manuscript's main idea.

Answer:

R- Lines 33 to 125 were restructured in this new version (marked in red). We shortened the introduction and the sections included in the paragraph “Characteristics of the virus” were omitted.

Question 4

The introduction and body of the review need to focus on describing whether or not the Antioxidant therapy could have a positive effect on COVID-19. First, the author needs to describe the role of ROS in the pathogenesis of SARS-CoV2. Then, whether antioxidant therapy could revert that. https://doi.org/10.1016/j.arcmed.2020.04.019, DOI: 10.23937/2474-3658/1510121,doi:10.1016/j.toxrep.2020.06.003,DOI: 10.3389/fcell.2020.00479

Answer

R- Thanks for the suggestions. In this new version in lines 235-261 were added to describe the role of ROS in the pathogenesis of COVID-19.

Question 5

There are several missing references and other references need to be changed for primary references (you need to go to the primary source of information). 

Answer

R- We revised the manuscript as suggested and the references were changed

Question 6

The figures should include controls to compare or other biochemical markers that validate that the reflected as Viral particles are viral particles.

Answer

R- We thank the reviewer for the observation. We do not have any biochemical marker or antibody against any protein of SARS-CoV-2 in our laboratory. However, the patient tested positive for the SARS-CoV-2 when the PCR test was applied and died with the diagonosis of COVID-19. The viral particles present in the biopsies of the representative photos taken with electron microscopy were measured and are in the range reported in the literature. In addition, they show the characteristic morphology of the virus as can be appreciated. Nevertheless, to make the suggested comparison and demonstrate that there are no viral particles in the tissues of subjects without COVID-19 infection, material from a patient with atrial fibrillation was processed and included as a positive control.

Question 7

All the clinical trials that are mentioned in the work must have the identification number obtained from Clinicaltrials.gov

Answer

Done, the identification number was added for the Clinical trials.

Question 8

All the tables need to be re-designed. Add divisions for each drug to follow the message easily and decrease the information burden. Also, add the references in numbers.

Answer

R- Done, the tables were restructured and the references were added.

Question 9

The authors need to re-write the abstract making clear to the main objective of the review. The same with the conclusion avoiding repeat information of the previous sections and focusing on the main importance of the work.

Answer

R- Done, the abstract was rewritten.

Other minor comments are shown in the following:

Question 10

Line 132: the authors refer to Spike-TMPRSS2 as 'It also binds to a host cellular trans-membrane serine protease''. This statement needs to be changed. TMPRSS2 is not a binding receptor of the viral spike. The viral spike after binding to ACE2 interacts with TMPRSS2 and is cleavage. Also, the citation should be hoffmann et al 2020 https://doi.org/10.1016/j.cell.2020.02.052 

Answer

R- Thanks you for your comment. This section was changed according to the information of the suggested manuscript and the reference was added in the reference section.

Question 11

Line 145:The ACE2 receptor-related signaling pathways play an important role in several pathologies that are considered as comorbidities that increase the death rate of COVID-19. There is a missing reference that supports this statement.

Answer

R- Done, the reference was added

Question 12

Line 146:The fact that the virus binds to this receptor and activates the signaling pathways might explain, in part, why symptoms are more pronounced in patients with CVD and related diseases including MS, diabetes,hypertension and hepatic diseases [15]. First, reference 15 did not reflect this statement. Second, the virus bind to ACE2 but is it related to activation of the intracellular signaling? If that is true, the author needs to show the respective reference that supports that. 

Answer

R- Done, this phrase was restructured.

Question 13

Line 156: COVID-2. Change to COVID-19

Answer

R- Done, this change was made.

Question 14

Sections COVID-19 and Diabetes/Brain did not apport any new knowledge, interpretation or support the main idea of the manuscript. I suggest either re-structure and adding more detail or delete. 

Answer

R- In this new version, the Diabetes/Brain sections were deleted.

Question 15

Line 221:COVID-19 severity is associated with liver diseases such as cirrhosis, non-alcoholic fatty liver disease, alcohol-related liver disease and chronic viral hepatitis[32]. Reference 32 did not reflect the author's statement. The author must re-write their statement or add the appropriated reference 

Answer

R- Done, the reference was changed

Question 16

Line 229: Reference 34 did not reflect the author's statement. The author must re-write their statement or add the appropriated reference 

Answer

R- Done, the reference was added.

Question 17

Line 231: There is also an elevation of gamma-glutamyl transferase which is a diagnostic biomarker for cholangiocyte injury, in 30 patients out of 56 (54%) with COVID-19 that required hospitalization. Reference???

Answer

R- Done the reference was added.

Question 18

Line 233: Reference 34 did not reflect the author's statement. The author must re-write their statement or add the appropriated reference 

Answer

R- Done the reference was added.

Question 19

Line 245,260 Reference 36 did not reflect the author's statement. The author must re-write their statement or add the appropriate reference. The presence of a receptor did not means infection. 

Answer

R- Thanks you for your comment, This observation is correct. Now this phrase was changed as can be observed in lines 176-178.

Question 20

Line 264: Reference 37 did not reflect the author's statement. The author must re-write their statement or add the appropriate reference. There is no direct relation to SARS-CoV2 in this reference.

Answer

R- Done, the reference was changed.

Question 21

Line 270: Check the title. There is not an inflammatory system. Also, this section needs to be deeply revised to show the importance related to the main idea of the review.

Answer

R- Done, it is correct. We apologize, the title was changed.

Question 22

Line 319 and 335: What is the importance of these sections to the main idea of the work? it should be deleted.

Answer

R- Done, in new version this sections were deleted.

Question 23

Line 428-441: This statement did not contribute to gain a deep understanding of HCQ therapy in SARS. It should be deleted. 

Answer

R- Done, in new version, this sections were deleted.

Question 24

The antiviral section needs to be restructured as well. The author mentioned some drugs as teicoplanin and carriomycin that are far to be used in antiviral therapy against SARS-CoV2. There several reviews in regard to clinical trial and SARS-CoV2. The author need to check that first. doi: 10.3390/ijms21072657. doi: 10.1128/AAC.00483-20 doi: 10.1002/phar.2398  doi: 10.21873/invivo.11949. doi: 10.1016/j.lfs.2020.117883 among other... If the authors want to include some information about drugs, should be related to the main candidates to be used in the therapy (actually in ongoing clinical trials). Carriomicin clinical trial started in February and it is not recruiting yet!

Answer

R- The reviewer is correct in his/her comment. Teicoplanin is an antibiotic and not an antiviral. We included it in this review due to its use as treatment against MERS-CoV and SARS CoV and in accordance to the study published by Barón et al who reported this fact. In the studies by Zhou N et al.  and Colson et al, it is reported that other authors have used Teicoplanin as treatment in other viral pandemias. Colson P et al suggested the possible use of this compound as treatment in COVID-19. The results of Wang Y et al report its use en ebola patients. However, since nowadays there are no proposed clinical trials for its use in this pandemia and more scientific evidence of its use is needed we agree in the possiblility of omiting this information in our review. Baron, S.A.; Devaux, C., Colson, P.; Raoult, D.; Rolain, J.M.; Teicoplanin: an alternative drug for the treatment of coronavirus COVID-19? Int. J. Antimicrob. Agents. 2020, 55, 105944. Zhou N, Pan T, Zhang J, Li Q, Zhang X, Bai C, et al. Glycopeptide antibiotics potently inhibit cathepsin L in the late endosome/lysosome and block the entry of Ebola virus, Middle East respiratory syndrome coronavirus (MERS-CoV),and severe acute respiratory syndrome coronavirus (SARS-CoV). J Biol Chem2016; 291:9218–32. doi:10.1074/jbc.M116.716100. Colson P, Raoult D. Fighting viruses with antibiotics: an overlooked path. Int J Antimicrob Agents 2016; 48:349–52. doi:10.1016/j.ijantimicag.2016.07.004. Wang Y, Cui R, Li G, Gao Q, Yuan S, Altmeyer R, et al. Teicoplanin inhibits Ebola pseudovirus infection in cell culture. Antiviral Res 2016;125:1–7. Varghese FS, Kaukinen P, Glasker S, Bespalov M, Hanski L, Wennerberg K, et al. Discovery of berberine, abamectin and ivermectin as antivirals against chikungunya and other alphaviruses. Antiviral Res 2016;126:117–24.

Regarding carryomicine, which is also an antibiotic that has been proposed as a treatment in clinical trials against COVID-19, the proposed study (ClinicalTrials.gov [Internet]. Bethesda (MD): National Library of Medicine (US), has not been started and there is no information on recruitment of patients. Therefore, we also agree in removing it from the present review.

Question 25

The section of drawbacks of combinatorial therapies needs to be carefully revised. This section focuses mainly on the side effects of protease inhibitors (lopinavir/rito). That is not a combinatorial therapy (lopinavir is a protease inhibitor and ritonavir is added to boost the effect of lopi by decreasing the metabolism. Thus, it is considered as a single drug)!!!! The author just described the interaction between other drugs and the antivirals. There are several examples of combinatorial therapy against SARS-CoV2 that are actually in clinical trials (more than 100 clinical trials registered in the NCBI) i.e: HCQ/camostat, lopi/rito/HCQ, Famotidine/HCQ, Arbidol/lopi/rito, Nitazoxanide/lopi etc.

Answer

R- We agree with the reviewer. Nevertheless we believe the doubt surged from the fact that the table containing the algorithm did not appear in the previous version of the manuscript. In this table that contains the algorithm, we explain the interactions that could exist when considering the standard therapies using antivirals, other drugs in combination with the use of antioxidants as coadyuvant. In the algorithm table we explain that for the selection of the use of antioxidants it should be considered that if lopinavir/ritonavir is being employed there could be interactions that should be analyzed before applying the antioxidant. We are sorry that the algorithm was not included in the last version and we will be careful that it is not omitted in this corrected new version.

Question 26

Line 812: Reference 137 did not reflect the author's statement. the authors need to go to the primary source.  

Answer

R- Done, the reference was changed.

Question 27

Line 822:QRC is safe for most people. However, there is no definitive evidence of its use in pregnancy. It is safe in doses of up to 1 gram daily, but it may induce side effects such as headache and paresthesias in the hands and legs, which are reversible. Safety is only affected with the use of high doses, due to the risk of renal toxicity. This statement needs to be re-write.. It is not clear the message.

Answer

R- Done, the phrase was rewriten.

Question 28

I am not a native speaker and I found several grammatical errors and in instances poor redaction. So, the language needs to be review by a native speaker to improve the quality and readability.

Answer

R- The manuscript was revised by a native English speaker.

Question 29

For this reviewer, the main idea of the review is interesting, but the manuscript is far to meet the quality criteria. The manuscript needs to be deeply revised, modified, and improved to be accepted for publication. 

Answer

The authors of the manuscript deeply appreciate the time that you invested and the observations made on our article.

Reviewer 2 Report

Thank you for the opportunity to review this manuscript in which the authors discuss  the  general  characteristics  of the  virus,  its  mechanism  of  action  and  the  way  in  which  the  mechanism  correlates  with  the comorbidities  that  increase  the  death  rate. The aim of the authors were to present  proposed  therapeutic measures  and  propose  the  use  of  antioxidant  drugs  to  help  COVID-19  patients.

Strength of the literature review is a very extensive database of current articles and a synthetic discussion of important issues related to COVID-19 treatment

Minor issues

  1. No abbreviations should be used in the abstract.
  2. It seems that this is not a systematic review of the literature but a narrative review of the literature selected by the authors. This should be included in the text. As well as in the text, authors should indicate the method used to find the items to review.
  3. The authors need to correct order of references
  4. In my opinion there are moderate English changes required

Best regards

Author Response

Referee 2

Thank you for the opportunity to review this manuscript in which the authors discuss  the  general  characteristics  of the  virus,  its  mechanism  of  action  and  the  way  in  which  the  mechanism  correlates  with  the comorbidities  that  increase  the  death  rate. The aim of the authors were to present  proposed  therapeutic measures  and  propose  the  use  of  antioxidant  drugs  to  help  COVID-19  patients.

Strength of the literature review is a very extensive database of current articles and a synthetic discussion of important issues related to COVID-19 treatment

Minor issues

Question 1

No abbreviations should be used in the abstract.

Answer

R- Done, the abbreviations were changed.

Question 2

It seems that this is not a systematic review of the literature but a narrative review of the literature selected by the authors. This should be included in the text. As well as in the text, authors should indicate the method used to find the items to review.

Answer

R- Done, thank you for your comment. It is in agreement with the observations made by referee 1 and therefore, this was changed as suggested.

Question 3

The authors need to correct order of references

Answer

R- Done, the order of the references was changed

Question 4

In my opinion there are moderate English changes required

Answer

R- The manuscript was revised arranged by a native English speaker.

The authors of the manuscript deeply appreciate the time you invested in reviewing the manuscript and the observations you made on our article.

Round 2

Reviewer 1 Report

The authors addressed most of the concerns. Now, the manuscript have
improved the quality and it warrants publication in Medicina.

Please change in the abstract lines 20 and 25

SARS-CoV-2 instead of Coronavirus-2. 

Author Response

20 Julio 2020

Marko Komlos

Assistant Editor

Medicina

We fully appreciate the time spent by the reviewers in evaluating our paper as well as their comments and advice.

I send the new version of the review paper "IS ANTIOXIDANT THERAPY A USEFUL COMPLEMENTARY MEASURE FOR COVID-19 TREATMENT? AN ALGORITHM FOR ITS APPLICATION" by the authors: María Elena Soto, Verónica Guarner-Lans Elizabeth Soria-Castro, Linaloe Manzano Pech and Israel Pérez-Torres. We have tried to follow their suggestions and enclose our replies, which are marked in red.

Best regards

Israel Pérez-Torres PhD.

Referee 1

Question 1

Please change in the abstract lines 20 and 25

SARS-CoV-2 instead of Coronavirus-2.

Answer

Done, was changed in the 20 and 25 lines coronavirus-2 by SARS-CoV-2

The authors of the manuscript deeply appreciate your time that you have invested and the observations made on our article.
